# Pulmonary Stretch and Lung Mechanotransduction: Implications for Progression in the Fibrotic Lung

**DOI:** 10.3390/ijms22126443

**Published:** 2021-06-16

**Authors:** Alessandro Marchioni, Roberto Tonelli, Stefania Cerri, Ivana Castaniere, Dario Andrisani, Filippo Gozzi, Giulia Bruzzi, Linda Manicardi, Antonio Moretti, Jacopo Demurtas, Serena Baroncini, Alessandro Andreani, Gaia Francesca Cappiello, Stefano Busani, Riccardo Fantini, Luca Tabbì, Anna Valeria Samarelli, Enrico Clini

**Affiliations:** 1Laboratory of Cell Therapies and Respiratory Medicine, Department of Medical and Surgical Sciences for Children & Adults, University Hospital of Modena and Reggio Emilia, 41125 Modena, Italy; marchioni.alessandro@unimore.it (A.M.); stefania.cerri@unimore.it (S.C.); ivana_castaniere@icloud.com (I.C.); darioandrisani@libero.it (D.A.); 72683@studenti.unimore.it (F.G.); giulibru92@gmail.com (G.B.); linda.manicardi3@gmail.com (L.M.); antomor93@hotmail.it (A.M.); annavaleria.samarelli@unimore.it (A.V.S.); enrico.clini@unimore.it (E.C.); 2University Hospital of Modena, Respiratory Diseases Unit, Department of Medical and Surgical Sciences, University of Modena Reggio Emilia, 41125 Modena, Italy; serena.baroncini@gmail.com (S.B.); alessandreani@yahoo.it (A.A.); gaia.cappiello@gmail.com (G.F.C.); fantini.riccardo@yahoo.it (R.F.); lucatabbi@gmail.com (L.T.); 3Clinical and Experimental Medicine PhD Program, University of Modena Reggio Emilia, 41125 Modena, Italy; 4Primary Care Department USL Toscana Sud Est-Grosseto, 58100 Grosseto, Italy; eritrox7@gmail.com; 5University Hospital of Modena, Anesthesiology Unit, University of Modena Reggio Emilia, 41124 Modena, Italy; stefano.busani@unimore.it

**Keywords:** mechanical ventilation, lung fibrosis, stress, strain, lung elastance, lung compliance, idiopathic pulmonary fibrosis, extra-cellular matrix, spontaneous breathing

## Abstract

Lung fibrosis results from the synergic interplay between regenerative deficits of the alveolar epithelium and dysregulated mechanisms of repair in response to alveolar and vascular damage, which is followed by progressive fibroblast and myofibroblast proliferation and excessive deposition of the extracellular matrix. The increased parenchymal stiffness of fibrotic lungs significantly affects respiratory mechanics, making the lung more fragile and prone to non-physiological stress during spontaneous breathing and mechanical ventilation. Given their parenchymal inhomogeneity, fibrotic lungs may display an anisotropic response to mechanical stresses with different regional deformations (micro-strain). This behavior is not described by the standard stress–strain curve but follows the mechano-elastic models of “squishy balls”, where the elastic limit can be reached due to the excessive deformation of parenchymal areas with normal elasticity that are surrounded by inelastic fibrous tissue or collapsed induration areas, which tend to protrude outside the fibrous ring. Increasing evidence has shown that non-physiological mechanical forces applied to fibrotic lungs with associated abnormal mechanotransduction could favor the progression of pulmonary fibrosis. With this review, we aim to summarize the state of the art on the relation between mechanical forces acting on the lung and biological response in pulmonary fibrosis, with a focus on the progression of damage in the fibrotic lung during spontaneous breathing and assisted ventilatory support.

## 1. Background

Mechanical homeostasis is defined as the capacity to generate and maintain a tissue mechanical environment to support organs’ physiological functions. This complex and dynamic process is the basis of the relationship between structure and function, and it is regulated at a subcellular level by actin cytoskeleton’s tension and integrin-mediated focal adhesion, directly interacting with external biophysical stimuli to trigger downstream cellular signals [1]. Physical forces (e.g., stress and strain) act on this biological system, affecting cell’s behavior and function, and finally resulting in tissue remodeling through cellular mechanotransduction, both during organ development and tissue damage [2,3]. The lung is an organ exposed to continuous mechanical stimuli during the respiratory cyclic stretch. The extracellular matrix (ECM) provides resident cells with a scaffold, acting as a mechanical support for the respiratory function; the ECM also plays a major role in transmitting external physical forces to cells. The influence by the lung’s mechanical load on the interaction between cells and the ECM contributes to a healthy mechanical tissue homeostasis [4]. However, the biological response to mechanotransduction could also affect the regenerative potential of the lung following an external insult, resulting in a pathological process [5]. As the lung is exposed to non-physiological stretch, especially when tissue injuries disrupt the mechanical homeostasis underlying normal tissue architecture and function, aberrant mechanisms of repair might be triggered. Pulmonary fibrosis can be considered as the final consequence of failure to restore physiological mechanical homeostasis caused by an abnormal tissue repair process driven by alveolar epithelium and leading to progressive fibroblast and myofibroblast proliferation and the excessive deposition of ECM. Furthermore, increasing evidence has shown that non-physiological mechanical load applied onto the lung and dysregulated mechanotransduction could both play an essential role in progression of lung fibrosis [6]. Idiopathic pulmonary fibrosis (IPF) is a chronic disease of unknown origin, which is characterized by progressive loss in lung function, culminating in respiratory failure and death [7,8]. In IPF, increased ECM stiffness and progressive scarring of lung tissue significantly change respiratory mechanics, making the lung more fragile and exposed to non-physiological stress during spontaneous breathing and mechanical ventilation, promoting lung damage and dysregulation of mechanotransduction and tissue repair [9]. The purpose of this review is to summarize the state of the art on the relation between mechanical forces acting on the lung and biological response in pulmonary fibrosis, with a focus on the progression of damage in the fibrotic lung during spontaneous breathing and assisted ventilatory support.

## 2. Matrix Abnormalities and Mechanical Behavior in Pulmonary Fibrosis

Lung ECM is organized into two basic compartments: basement membranes, which are thin sheets of glycoproteins covering the basal side of epithelia and endothelia, and interstitial matrices, which form a loose and fibril-like meshwork that interconnects structural cell types within tissues and provides mechanical stability and elastic recoil of the lung [10]. The biomechanical features of the lung are the result of ECM composition, and they depend on a complex network of fibrous proteins, glycoproteins, proteoglycans, and associated modifying molecules (e.g., metalloproteases, matricellular proteins) that represent the non-cellular portion of lung structure, which is also called the “*matrisome*” [11]. *Matrisome,* the overarching architecture of the lung, mainly consists of fibrillar collagens (types I, II, III, V, and XI) and elastic fibers, which exhibit different mechanical properties and represent the main stress-bearing constituents of lung tissue. Indeed, fibrillar proteins such as collagens are characterized by great tensile strength but low elasticity, while elastic fibers, mainly composed of elastin, show high elasticity and low tensile strength. Therefore, the quantitative and architectural changes of these components can influence the elastic return of the lung and the nonlinear stress/strain characteristics during breath. In particular, collagen fibers, which are folded in the resting position, are stretched only at high pulmonary volumes close to total lung capacity, and they act as a blocking system determining both a limitation in lung distention and the origin of the curvilinear stress–strain behavior, whereas elastin molecules account for the lung elastic recoil [12]. The ECM in fibrotic lungs shows different mechanical properties and biochemical composition to healthy lungs. The excessive deposition of fibrillar collagen, predominantly around myofibroblasts, is the key feature of architectural derangement in IPF, and it results in stiffness of the area within fibroblastic foci [13]. Some studies have described a significant change in collagen composition in IPF, with an increase in the amount of type I and type V collagen and a decrease in the amount of type III collagens [14]. However, an increase in the expression of proteoglycans such as versican and decorin, and glycoproteins such as fibronectin, have also been reported in experimental models of lung fibrosis [15]. Furthermore, studies on animal models of bleomycin pulmonary fibrosis and on IPF lungs have also shown an increasing expression of elastin in the fibrotic area, suggesting a process of “fibroelastosis” rather than an exclusive process of fibrosis in acute and chronic idiopathic interstitial pneumonia [16,17]. Recent evidence suggests that ECM composition not only defines the tissue architecture of the lung, but it is also important in promoting fibrotic changes and disease progression in fibrotic lung. Elastin and glycoprotein such as fibronectin are both essential in driving cells toward a profibrotic phenotype with the induction of myofibroblast differentiation through transforming growth factor β1 (TGF-β1) pathway amplification [18]. Some experimental evidence shows that when fibroblasts are cultured on a stiffer IPF matrix, they develop an activated myofibroblast phenotype with typical features reported in IPF disease [19]. However, in IPF lungs, fibrosis areas and stiffer matrix distribution are uneven, with histological and radiological appearance that has been named usual interstitial pneumonia (UIP). The key of the UIP pattern is the distribution of fibrosis that mainly involves the lower lobes with subpleural accentuation. In these regions, areas of airspace enlargement and fibrotic retraction, namely honeycombing, are a marker of the advanced stage of the disease. Moreover, the microscopical appearance is characterized by spatial and temporal heterogeneity, the latter resulting in a patchy fibrotic reaction with prevalent involvement of the peripheral area of the secondary pulmonary lobule, while the central portion is often spared [20]. Studies that analyzed IPF lung with electron microscopy have shown a loss of type 1 alveolar epithelium, denudation of alveolar basal lamina, and subsequent incorporation of denuded basal lamina into the interstitial alveolar septum. This process has been named “collapse induration” and is associated with the permanent obliteration of an alveolar unit and impaired lung mechanics. Based on these assumptions as a whole, the fibrotic lung is characterized by a highly inhomogeneous mechanical microenvironment, where lung areas with preserved elasticity are contiguous to areas of rigid lung, resulting in a peculiar mechanical behavior during inflation, especially in terms of the stress/strain relationship.

## 3. Stress and Strain Behavior of the Normal and Pathologic Lung

Stress is defined as the magnitude and direction of forces acting on a body, while strain is the magnitude and direction of the consequent deformations [21]. Elastic (Young’s) modulus is a measure of the ability of a material to change in length when stretched or compressed, and it is defined as stress/strain relationship. Biological networks show highly nonlinear stress–strain behavior and are characterized by tissues that stiffen up when exposed to increasing deformation. High values of Young’s elastic modulus (*E_Y_*) lead to lower values of extensibility and of elongation at break (*λ*_max_). The relationship between stress (*σ*), deformation ratio (*λ*, defined as the ratio of the sample’s instantaneous size to its initial size), and *E_Y_* is described as a constitutive equation for all equilibrium network deformations, and it has been validated for a broad range of complex networks:$$\sigma true\left(\lambda \right)=\frac{Ey}{9}\left(\lambda^{2}-\lambda^{-1}\right)\left[1+2{\left(1-\frac{\beta \left(\lambda^{2}+2\lambda^{-1}\right)}{3}\right)}^{-2}\right]$$
in which *β* is the strand–extension ratio [22]. Tissue and organs possess different *E_Y_* elastic moduli depending on ECM composition, fat content, cell types, and the degree of mineralization, and they range from extremely low value (blood plasma, about 50 Pa) to very high values in stiff tissues (bone, about 100,000 kPa) [11]. Different biomechanical models consider the lung as a homogeneous elastic continuum undergoing only small distortions from a state of uniform inflation, whose behavior can be described with different elastic moduli: *E_Y_*, describing tensile elasticity, bulk modulus (K), describing a uniform inflation, and shear modulus (μ), describing an isovolumetric deformation. Normal human lung parenchyma has a mean *E_Y_* of 1.96 kPa, but this value depends on the assessed lung region [23].

Fibrotic lung diseases are characterized by increased lung stiffness and show significantly higher *E_Y_* compared to the healthy lung. In measurements performed in IPF lungs, the lung elastic modulus displayed a bimodal distribution of stiffness with an *E_Y_* average of 16.55 kPa, but with a wide variability. The different zonal elasticity in the IPF lung is related to the spatial and temporal heterogeneity of the areas of fibrosis that are located close to the areas of the spared lung. This elastic patchwork has profound implications on the stress–strain curve and on the consequences of the mechanical load exerted on the lung scaffold, both in spontaneous breathing and during mechanical ventilation.

In engineering terms, the behavior of a solid body under the action of a physical force can be described using a stress–strain curve of the material (Figure 1). A *stress–strain* curve consists of an initial part in which the relationship between the two elements is linear (elastic area), which is represented by the constitutive equation of an elastic solid (Hooke’s Law):*σ* = *E_Y_* × *Strain*.B.

The elastic limit on a *stress–strain* curve is the point at which the behavior of the material switches from elastic to plastic (Figure 1). When the elastic limit is exceeded, the plastic strain cannot recover when the load is removed and will remain as permanent set.

Studies in animal models show that the respiratory system presents a viscoelastic body behavior that fulfills the constitutive equation of a solid (Hooke’s law) [24].

In physiological terms, the equation “B” can be written as follows:$$P_{TP}=E_{Y}\times \frac{VT}{FRC}$$
in which *P_TP_* is the transpulmonary pressure (or retraction pressure of the lung), which is defined by the difference between alveolar pressure (*P_alv_*) and pleural pressure (*P_pl_*). P_TP_ is the force that causes lung inflation, and that can be referred to as the stress applied to the lung. *VT* refers to tidal volume (volume of air moved during each ventilation cycle), and *FRC* is functional residual capacity (that corresponds to the volume of air at the end of expiration). Therefore, the relationship between these two parameters (*VT/FRC*) represents the pulmonary deformation during the respiratory cycle and can be defined as the lung strain. *E_Y_*, in terms of respiratory physiology, coincides with *specific elastance* of the lung, which corresponds to the value of *P_TP_* that is needed to double the volume of FRC. The value of specific elastance in a healthy human lung is 13.5 cmH_2_O, and it does not seem to change with age [25,26]. In animal models of mechanical ventilation, the linear relationship between stress and strain is preserved for a strain value less than 1 (elastic area), while for strain values between 1.5 and 2, the relationship loses its linearity until the elastic limit is reached and lung injury begins to occur [24]. Furthermore, in mechanically ventilated patients with ARDS, increased strain is associated with alveolar proinflammatory response that could be explained by different mechanisms, among which a mechanotransduction process activated beyond a certain threshold of lung deformation may have a major role [27].

## 4. The Micro-Strain Concept in the Fibrotic Lung

In the fibrotic lung, the *specific elastance* of the lung is higher than normal, and the stress/strain curve reaches the elastic limit at lower stress and strain values than in healthy lung (Figure 1). Moreover, the large regional distortion during inflation to which the lung is subjected can allow the dysregulation of mechanotransduction and the occurrence of microdamage of the epithelial and pulmonary scaffold even at low levels of global strain. Part of this phenomenon can be explained by the regional differences between the isotropic expansion of the lung (healthy lung) and the anisotropic behavior (fibrotic lung) during inflation, introducing the concept of micro-strain. For isotropic homogeneous materials, such as healthy lung, uniform expansion could simply be described by *K*, while non-uniform distortion could be described by *μ* both related to *E_Y_* and Poisson’s ratio (*v*) through the following equation:$$K=\frac{E_{y}}{3\left(1-2V\right)}\;\;\;\;\;\;\;\;\;\;\;\;\mu =\frac{E_{y}}{2\left(1+v\right)}$$

In this situation, the lung parenchyma is usually modeled by elastic moduli that are solely functions of the transpulmonary pressure. Indeed, the experimental measurement is approximately dependent upon *P_TP_*:$$\mu =\alpha P_{TP}$$
where *α* is the constant of proportionality.

Based on these laws, the healthy lung that is exposed to physiological inflation is on the elastic part of the stress–strain curve, where the linear relationship makes the load to which the lung is exposed predictable, and the physical forces are distributed homogeneously in the different lung regions [28]. In pulmonary fibrosis, the parenchyma becomes physically inhomogeneous, and its properties become globally anisotropic. If deformations are large during lung inflation, the linear theory becomes invalid, and the elastic moduli cannot be regarded as constant for a given *P_TP_*. The distortion of the parenchyma causes a re-orientation and an associated change in the strain of the load-bearing components; as a result, moduli are functions not only of *P_TP_* but also of the type and magnitude of the distortion [29]. In particular, during an inspiratory effort, the IPF lung may exhibit anisotropic behavior with different regional deformations not described by the standard stress–strain curve. Especially in the interface regions between fibrotic and spared lung areas, the elastic limit can be reached due to the excessive deformation of parenchymal areas with normal elasticity, which are surrounded by inelastic fibrous tissue or collapsed induration areas that tend to protrude outside the fibrous ring, with consequent lung damage. The effect observed in some lung areas is similar to that shown in stress balls called “squishy balls” and can be amplified by mechanical ventilation (Figure 1) [30]. Therefore, in the fibrotic lung, macro-strain is not descriptive of regional micro-strain, even when the analysis of the global stress/strain curve of the whole lung appears to be in the elastic area (Figure 1). Micro-strain could act, during lung inflation, as a mechanical stimulus on the ECM, which in turn is transmitted to alveolar cells, by activating mechanotransduction. Furthermore, in highly inhomogeneous fibrotic lungs, this mechanical phenomenon may promote cyclic tractional epithelial injury in adjacent lung tissue during spontaneous breathing with persistent damage and repair that finally results in fibrosis progression. Although no data are available to describe the extent of lung strain needed to cause pulmonary fibrosis, Albert et al. suggested that the relationship between strain and fibrosis could be represented as follows: pulmonary fibrosis ≈ strain × f × t
where strain is represented by the deformation above the threshold that initiates mechanotransduction, while f and t represent the frequency and duration, respectively, over which strain is applied [31]. Moreover, the honeycomb appearance in end-stage disease could be explained by a persistent “traction-like phenomenon” causing micro-strain to occur around the area of dense fibrosis or “collapse induration”. Indeed, the mainly basal distribution of honeycombing observed in pulmonary fibrosis could also be the result of the increased *P_TP_* swings that occur in this region during ventilation, which promote recurrent tractional injury [32,33,34]. The micro-strain concept and the regional “squishy ball” phenomenon can also explain mechanical ventilation damage that may occur during protective mechanical ventilation. Figure 1 describes the relationship between global stress/strain curve and regional micro-strain in the fibrotic lung.

## 5. The Mechanotransduction Process: Biological Response to Stretch and Progression in the Fibrotic Lung

Cells respond to mechanical forces in their environment through cell division, differentiation, and migration, as well as promoting morphogenesis and tissue repair. During the respiratory cycle, stress and strain act on the ECM; then, from the ECM, these physical signals are transferred to the cells, promoting specific biological functions of alveolar cells in healthy subjects. It is widely believed that the stretch of alveolar type II cells, occurring during breathing, is one of the main triggers for surfactant release [35]. The most relevant mediators involved in transducing signals from the biomechanical environment to intracellular pathways include integrins, growth factor receptors, G-protein-coupled receptors, mechanoresponsive ion channels (e.g., Ca^2+^), and cytoskeletal strain responses [36]. Integrins are part of a large family of heterodimeric transmembrane receptor proteins. They consist of two domains: a cytoplasmic domain, linked to the actin cytoskeleton, and an extracellular domain that recognizes specific polypeptide sequences in the ECM molecules. The transduction of mechanical forces is modulated through mechanosensitive focal adhesion proteins, a complex macromolecular structure consisting of scaffolding, docking, and intracellular signaling proteins that collectively serve as an interface between integrins and actin cytoskeleton [37]. Integrin receptors, once activated, can modulate the recruitment of focal adhesion proteins, promote actin polymerization, enhance adhesion stability and contractility, as well as activate mechanosensitive genes. Focal adhesion kinase (FAK) is one of the first molecules recruited in intracellular mechanotransduction; the autophosphorylation of FAK leads to its activation and a series of downstream signals within the cytoplasm. Other cytoplasmatic proteins take part in the integrin adhesion process by linking the cytoskeleton to the ECM. Talin is a 270 kDa protein consisting of three components: a N-terminal globular head that interacts with the cytoplasmatic domain of integrin, a rod domain containing several binding sites for vinculin, and a C-terminal helical domain. Vinculin is a multidomain cytoplasmatic protein that is able to interact with other adhesion complex proteins, and it acts as the main partner of talin in the mechanosensing process. Cooperating with talin, vinculin links integrin to actin cytoskeleton. In the absence of mechanical strain, the talin rod remains fully structured, and no vinculin binding sites are exposed. On the other hand, once a force is applied, the amount of vinculin sites available for binding depends on the magnitude of the physical force itself. The talin–vinculin mechanosensitive cooperation is considered essential to stabilize the talin–F-actin interaction and to transmit the magnitude of the signal down intracellular pathways [38]. Although the molecular process by which mechanical signals are conveyed from the periphery to the cell nucleus has not been fully understood, some studies suggest that the nucleus has a mechanosensitive apparatus of its own and that the cytoskeleton can transmit forces across the nuclear envelope, altering the nuclear environment and promoting genes transcription [39]. Nuclear envelope spectrin-repeat proteins (Nesprin) are a super family of proteins located into the nuclear envelope that establish nuclear–cytoskeleton connections. Nesprin contains an evolutionary conserved c-terminal Klarsicht ANC-1 and Syne Homology (KASH) transmembrane domain; it binds to the C-terminal domain of the trimeric SUN protein that interacts with the nuclear lamina. Nuclear lamina structures provide the internal surface of the nuclear envelope with a scaffold and link the cytoskeleton to the nucleoskeleton through SUN–KASH bridges and the lamina protein Emerin [40,41].

In conditions of excessive lung stretch, animal models have shown different signaling pathways involved in the induction of pulmonary fibrosis through mechanical transduction, including Rho/rho associated protein kinase (ROCK), myocardin-related transcription factor-A (MRTF-A), and yes-associated protein 1 (YAP)/(transcriptional coactivator with PDZ-binding motif) TAZ signaling pathways [6]. Rho GTPases (Rac, Rho, and CDC42) are signaling G proteins that are distributed across the lower surface of the cell and regulate cytoskeletal dynamics by controlling actin polymerization and myosin II-mediated contraction. One of the main biological actions of Rho is achieved by its effector Rho-associated protein kinase (ROCK, presenting in two isoforms ROCK1 and ROCK2). The Rho/ROCK regulation of actin cytoskeletal dynamics acts as a trigger of the activity and the nuclear localization of myocardin-related transcription factor (MRTF) and the Hippo pathway effector YES-associated protein (YAP) and its paralog transcriptional coactivator with PDZ-binding motif (TAZ) [42,43]. Some studies suggest that Rho/ROCK activity plays a crucial role in the development and progression of pulmonary fibrosis. Observations from experimental mice models of lung fibrosis and from human subjects with IPF suggest that the activation of the Rho/ROCK pathway sustains progressive fibrotic disorders [44]. The blockade of the Rho/ROCK pathway prevents lung fibroblast differentiation into myofibroblasts and inhibits the development of pulmonary fibrosis secondary to lung injury in vivo. Furthermore, in animal models, ROCK1- and ROCK2-haploinsufficient mice exhibited protection from bleomycin-induced pulmonary fibrosis [44,45,46,47]. Rho-mediated actin polymerization results in MRTF-A (also known as MKL-1) nuclear translocation, activation of α-smooth muscle actin (α-SMA) gene expression, and type I collagen synthesis [48]. In particular, in the nuclear environment, MRTF interacts with the serum response factor (SRF), resulting in enhanced transcription via conserved CArG box DNA element of type I collagen α2 chain (COL1A2) gene promoter. SRF/MRTF signaling is considered as having a central role in the differentiation of fibroblasts into the myofibroblasts phenotype. Both molecular and mechanical factors associated with fibrosis, including TGF-β1 and exposure to stiff substrates, have been found to promote MRTF-A nuclear translocation in fibroblasts. Furthermore, the blockade of the nuclear action of SRF/MRTF attenuates myofibroblast’s differentiation and enhances myofibroblast’s susceptibility to apoptosis in vitro [49]. Rho/ROCK activity is also required for YAP/TAZ nuclear translocation to allow its different biological functions such as the modulation of genes transcription. YAP/TAZ are central transcriptional coactivators, interacting with DNA-binding transcription factors to regulate targeted gene expression; being coactivators, they are in fact unable to directly bind to DNA [50]. In mammalians, TEAD family transcription factors (TEAD 1–4) are the main partners of YAP in driving gene transcription, while the role of other transcriptional factors, including Smad, runt-related transcription factor1/2 (RUNX1/2), p63/p73, and ErbB4, is less clear [51]. Recently, some studies have shown that mechanical signals arising from the cells’ microenvironment control both the localization and activity of YAP/TAZ. High mechanical stress induced by cell spreading or by culture on a hard surface promotes TAZ/YAP nuclear accumulation and mechanoactivation [52]. Specifically, TAZ/YAP nuclear accumulation is controlled by cell shape, by the rigidity and topology of the ECM, and by shear stress. YAP and TAZ are usually located in the cell cytoplasm of cells that receive low levels of mechanical signaling (cells attached to a soft ECM); conversely, they are accumulated in the nucleus of cells exposed to high mechanical stress or experiencing deformation and cytoskeletal tension. YAP/TAZ exerts its profibrotic effect through interaction with nuclear transcriptional factors and the activation of different ECM genes including plasminogen activator inhibitor-1 (PAI-1), connective tissue growth factor (CTGF), and COL1A1 and COL1A2. Some studies have demonstrated the presence of nuclear YAP accumulation and YAP targeted gene expression in lung fibroblasts grown on stiff matrices and in IPF epithelial cells, suggesting the ability of YAP/TAZ activation to drive profibrotic response in the lung [53,54]. Furthermore, nuclear exclusion and the subsequent suppression of YAP/TAZ gene transcription could inhibit the profibrotic activity induced by TGF-β in fibroblasts in vitro [55,56]. Figure 2 summarizes the mechanotransduction pathways involved in the progression and development of lung fibrosis currently identified in animal models.

## 6. Role of Alveolar Type 2 Cells in the Progression of Lung Fibrosis

Alveolar type II (AT2) cells are considered as the main cell type for the synthesis and secretion of lung surfactant. The pulmonary surfactant appears to play a key role in the evolution of acute respiratory disease toward fibrosis. Abnormalities in the synthesis and function of surfactant, especially of its structural proteins, surfactant protein A (SP-A) and surfactant protein D (SP-D), both acquired and inherited, are associated with a higher prevalence of the development of pulmonary fibrosis compared to healthy volunteers without protein alterations [57,58]. Additionally, several studies also described the association between genetic polymorphisms for surfactant proteins and ARDS [59]. A rare mutation of the gene encoding surfactant protein A2 (SP-A2) is associated with the development of familial idiopathic pulmonary fibrosis [60]. Furthermore, several Gram-negative infections seem to alter the endogenous surfactant by inhibiting the biosynthesis of phospholipids and/or its biophysical function with the secretion of elastase, which favors the path toward fibrosis [61,62].

Focusing on the role of AT2 cells apart from their contribution to the genesis of surfactant, it has recently been shown that AT2 cells also function as alveolar stem cells in the lungs, being able to self-renew and differentiate into alveolar type I (AT1) cells, which are the main epithelial component of the alveolar–capillary barrier in gas exchange—hence their key role in regeneration and repair after lung injury [63]. Recent evidence suggests that AT2 cell depletion and/or dysfunction could play a relevant role in the pathogenesis and progression of lung fibrosis [64]. Multiple elements, including genetic, environmental, stress, and age-related factors, could contribute to AT2 cell dysfunction and loss of their homeostatic role in IPF. A prevailing concept is that AT2 cell depletion due to repeated micro-injury in the alveolar epithelium could constitute an early event in IPF pathogenesis. Furthermore, AT2 apoptosis correlates with severe or prolonged endoplasmic reticulum (ER) stress, which is in turn associated with the development of fibrotic disorders in multiple organs [65]. Susceptibility to ER stress is also associated with different factors, such as smoking and aging, which are linked with IPF development [65,66,67]. However, in addition to the depletion of epithelial cells, in pulmonary fibrosis, AT2 cell dysfunction could alter their ability to repair alveolar tissue after chronic micro-injury, resulting in a modification of mechanical homeostasis. Indeed, some observations have shown that in fibroblastic foci, AT2 cells have an impaired renewal capacity and could promote fibrogenesis with profibrotic factors production [68,69]. Recent studies show that in AT2 cells, YAP is a key mediator in regulating mechanical tension-induced alveolar regeneration in response to lung injury through the activation of Cdc42/F-actin/mitogen-activated protein kinase (MAPK)/YAP signaling cascade [70]. Cell division control protein 42 homolog, also known as Cdc42, is a small GTPase protein that is part of the Rho family, and whose function is involved in control c-Jun N-terminal kinases (JNK)-p38 kinases activation and phosphatidylinositol 4,5-bisphosphate (PIP2)-induced actin assembly [71]. The Cdc42 pathway is also involved in premature epithelial cellular senescence, which is a cellular process linked to IPF [72].

Pneumonectomy is one of the main models employed to study mechanical forces driving alveolar regeneration. Indeed, the loss of alveoli after PNX causes significantly increased mechanical tension on the remaining alveolar epithelium, while an efficient alveolar regeneration response is able to reduce the intensity of the mechanical tension to which the alveoli are exposed. It has been shown that Cdc42-controlled actin remodeling is required for JNK and p38 activation and nuclear YAP expression in AT2 cells during PNX-induced alveolar regeneration [70]. Moreover, a recent study has shown that AT2 cells that lack Cdc42 function are not able to differentiate into AT1 cells in PNX-treated or aged mice, and therefore, they are unable to regenerate new alveoli after lung injury, resulting in increased mechanical tension on the alveolar epithelium [73]. Furthermore, in Cdc42-null mice, the loss of Cdc42 functions in AT2 cells leads to progressive lung fibrosis in post-PNX lungs, with a pattern similar to IPF (from the periphery to the center of the lung). Fibrotic development was reduced in CdC42-null mice when mechanical tension was released. It is known that the application of mechanical stretch on AT2 cells is able to activate the TGF-β pathway, which induces disturbance of the homeostatic microenvironment, leading to an aberrant wound healing promoting the fibrotic process. In an ex vivo model, mechanical tissue stretch induces the activation of TGF-β_1_ signals through Rho/ROCK and αv integrins interactions [74]. Wu et al. suggested that an increased mechanical tension caused by defective AT2 cells alveolar renewal capacity and associated with tissue stretch occurring during spontaneous breathing could cause an aberrant TGF-β signaling loop activation resulting in fibrosis progression [73]. Mechanical stress is heterogeneously distributed during lung inflation with the posterior bases of the lower lobes being the sites where the magnitude of transpulmonary swing is more evident. With this assumption, the progression of lung fibrosis could start in these regions, progressively extending in a caudal-cranial mode along the axis of distribution of mechanical forces. We can speculate that in the lung with IPF, the micro-strain due to the regional squishy ball behavior can behave as a catalyst of the profibrotic mechanotransduction pathways in a spatial manner, facilitating the progression of fibrosis from the periphery of the lung toward the center. Furthermore, this coupling of pulmonary stretch and profibrotic pathways can be partly promoted by the altered mechanical homeostasis due to the dysfunction of AT2 cells that are unable to renew damaged alveolar tissue.

## 7. Conclusions and Clinical Implications

Mechanical stresses might be involved in triggering and promoting the dysregulation of the key molecular pathways governing lung tissue repair, thus leading to fibrotic changes. In this complex scenario, the presence of established fibrosis may enhance the impact of lung stretch during both spontaneous breathing and mechanical ventilation. Several factors may influence the relationship between alveolar architecture and the response to the physical stimuli applied to the lung. Structural changes occurring in the respiratory system with advancing age, such as the decrease of lung elasticity and the stiffness of the chest wall, may predispose the aging lung to harmful responses once subjected to stress. In this line, cellular senescence leads to replicative arrest, apoptosis resistance, and the acquisition of a senescence-associated secretory phenotype, which involves the release of several inflammatory, growth-regulating, and tissue-remodeling factors and could thus contribute to pro-fibrotic responses [75]. Furthermore, several forms of acute lung injury (namely ARDS) can amplify the mechanical stresses to which the lung is subjected. In particular, parenchymal inhomogeneity can act as a stress raiser at the interface of regions characterized by different tissue elasticity, thus enhancing lung damage and abnormal repair response. The activation of the aforementioned pathways of mechanotransduction could be implied in the onset and progression of lung fibrosis related to acute lung injury.

Interstitial lung diseases (not limited to IPF) showing a UIP pattern are subjected to acute exacerbations with dramatic gas exchange impairment requiring ventilatory assistance [76]. Once mechanical ventilation is needed, a protective strategy is advisable in order to reduce lung stretch and consequently avoid fibrotic lung damage progression via mechanotransduction [9]. Differently from the recommended ventilatory management of ARDS patients, an open lung approach with a high level of PEEP to prevent atelectotrauma should be rather avoided. Indeed, it is even arguable that elevated PEEP values are able to contrast the alveolar recruitment–derecruitment phenomenon, this happening at the cost of a remarkable lung parenchymal stress. Although data regarding the mechanical behavior of fibrotic lungs exposed to Ptp-titrated PEEP are lacking, a preliminary report on five patients with AE-ILD undergoing MV showed a remarkable (or significant, to avoid repetitions) mechanical disadvantage [30]. These physiological changes may be explained by the mechano-elastic model of the lung acting as a “squishy-ball”, in which the lung is represented by a patchwork of extremely different elasticities arranged contiguously. In this scenario, the concept of micro-strain does not allow managing the ventilatory assistance of fibrotic lungs trusting the classical correspondence between the stress–strain curve and the ventilatory parameters recommended to protect the lung (i.e., VT < 6 mL/kg, plateau pressure < 25 cmH_2_O, driving pressure < 15 cmH_2_O). Therefore, mechanical ventilation might promote fibrosis progression even when the principles of protective mechanical ventilation strategy are rigorously followed. The pathophysiology described above leads to speculation on whether the forces acting on the lung during spontaneous breathing (i.e., vigorous inspiratory effort) may determine a local excessive stretch, thus favoring a feed-forward loop of fibrosis. If this is true, elevated transpulmonary pressure swings such as those produced by maximal physical exercise or large breaths should be avoided as potentially harmful, particularly in patients with advanced disease. In this line, a tailored approach to pulmonary rehabilitation for patients with IPF should be encouraged, taking into account the mechanical properties of the lung exposed to physical stretch. Further research in this field is required to clarify the complex interaction between mechanical stressors and lung response in patients with fibrotic lung disease and UIP pattern.

## Figures and Tables

**Figure 1 ijms-22-06443-f001:**
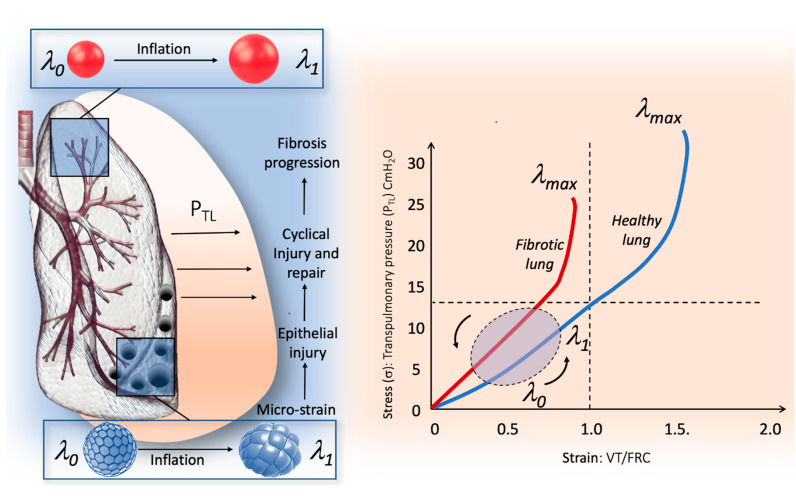
Relationship between micro-strain and global stress–strain curve in human lung. *The right part* of the figure illustrates the stress–strain curve in the healthy lung (blue line) and in the fibrotic lung (red line). In the fibrotic lung, the stress–strain is steeper compared to healthy lung, which is due to the higher specific elastance (the slope of the curve in its linear portion); therefore, the transition from elastic to plastic behavior is achieved for lower stress–strain values. *The left part* of the figure illustrates the behaviors of the IPF lung during inflation, from l0, which correspond to the elastic equilibrium of the respiratory system (i.e., functional residual capacity) to 11, which corresponds to the end of tidal volume. The fibrotic lung is made up of a patchwork of areas of different elasticity predominantly in the basal and subpleural zone, in which areas of dense fibrosis and areas of spared lung tissue are contiguous. During inflation, areas of the lung with normal elasticity surrounded by inelastic tissue protrude outside the lung surface exhibiting squishy ball-like behavior. Thus, in these areas, the global lung strain does not represent the micro-strain. The traction exerted by the non-physiological cyclic micro-strain could result in epithelial injury and finally in fibrosis progression.

**Figure 2 ijms-22-06443-f002:**
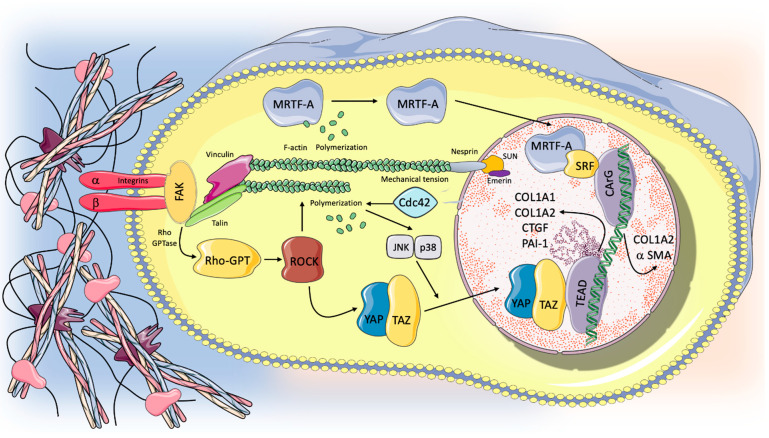
Mechanotransduction and intracellular pathways potentially involved in pulmonary fibrosis progression. Unphysiological mechanical stimuli act on the ECM, which in turn transmitted the physical force to integrins at the cells’ surface. Integrin clustering, through mechanosensitive focal adhesion proteins (i.e., FAK, talin, vinculin), promotes actin polymerization and cytoskeletal remodeling that can transmit forces across the nuclear envelope through specialized proteins (Nespirn, SUN, emerin), potentially influencing gene transcription. Furthermore, different intracellular pathways implicated in the induction of pulmonary fibrosis could be activated via mechanotransduction process. Rho through its effector ROCK acts as a trigger for the activity and nuclear localization of MRTF and YAP-TAZ, which in turn activated the transcriptions of profibrotic genes (COL1A1, COL1A2, CTGF, PAI-1, a SMA). Cdc-42, a GTPase protein that is part of the Rho family, is involved in alveolar regeneration after it increased mechanical tension in response to lung injury via JNK/p38 activation and nuclear YAP expression in AT2 cells. Therefore, the loss of Cdc-42 function could result in a dysfunction in alveolar renewal after lung damage. For further details, see the text.

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
