# Peer review of "Pulmonary Stretch and Lung Mechanotransduction: Implications for Progression in the Fibrotic Lung"

_ijms, 2021, doi:10.3390/ijms22126443_

Round 1
Reviewer 1 Report
I appreciate the effort of the authors.This revew is sufficient to reach the conclusion described in this manuscript at this time.Figure 2 is very easy to understand, but another detailed diagram will help you understand it.
Author Response
We really thank the Reviewer for the careful reviewing of our work. We are also grateful for the appreciation of our review. We have carefully considered the Reviewer's suggestion regarding Figure 2. However we do believe that the insertion of a new diagram might result redundant in describing the related molecular pathways. Thus we leave the decision as to whether proceeding or not to duplicate Figure 2 in a detailed diagram to the Editors. Best regards.Reviewer 2 Report
In this paper, the author summarized the relation between mechanical forces acting on the lung and biological response in pulmonary fibrosis, with a focus on the progression of damage in the fibrotic lung during spontaneous breathing and assisted ventilatory support, which provided a comprehensive description of the relation between Pulmonary stretch and lung mechanotransduction and progression of lung fibrosis to the research communities.
In this review paper, I have few concerns:
The author described the relation between “pulmonary stretch and lung mechanotransduction” and lung fibrosis, which is good. I am concerned that what can induce the “Pulmonary stretch and lung mechanotransduction”, internal factors (such as diseases, aging) or external factors (such as bad behaviors, injuries), the author did not mention that, which I think should be discussed in this review paper.
Another concern is that the author describes the context in an engineering way that introduces several formulae, I think the author needs to add some contents from the biological side, such as what biological pathways are affected during the progression of lung fibrosis because of the pulmonary stretch and lung mechanotransduction.
Minor:
- In line 44, there is an ‘at’, please check and correct.
- In line 163, Hook’s law did not show properly, please correct.
- In line 136, 137,148, 214, 221, what is that symbol: , please check throughout you whole text
Author Response
We would like to thank the Reviewer very much for the thoughtful and constructive review of our paper. We have carefully read your comments and we have tried to modify the revised manuscript accordingly. Comment 1 The author described the relation between “pulmonary stretch and lung mechanotransduction” and lung fibrosis, which is good. I am concerned that what can induce the “Pulmonary stretch and lung mechanotransduction”, internal factors (such as diseases, aging) or external factors (such as bad behaviors, injuries), the author did not mention that, which I think should be discussed in this review paper. Answer to comment 1 We thank the Reviewer for this comment. We have added a paragraph in the final part of the manuscript in order to discuss the suggested issue. "Mechanical stresses might be involved in triggering and promoting dysregulation of the key molecular pathways governing lung tissue repair, thus leading to fibrotic changes. In this complex scenario, the presence of established fibrosis may enhance the impact of lung stretch during both spontaneous breathing and mechanical ventilation. Several factors may influence the relationship between alveolar architecture and the response to the physical stimuli applied to the lung. Structural changes occurring in respiratory system with advancing age, such as the decrease of lung elasticity and the stiffness of the chest wall, may predispose the ageing lung to harmful responses once subjected to stress. In this line, cellular senescence leads to replicative arrest, apoptosis resistance, and the acquisition of a senescence-associated secretory phenotype, that involves the release of several inflammatory, growth-regulating and tissue-remodeling factors and could thus contribute to pro-fibrotic responses [75]. Furthermore, several form of acute lung injury (namely ARDS) can amplify the mechanical stresses to which the lung is subjected. In particular, parenchymal inhomogeneity can act as stress raiser at the interface of regions characterized by different tissue elasticity, thus enhancing lung damage and abnormal repair response. The activation of the aforementioned pathways of mechanotransduction could be implied in the onset and progression of lung fibrosis related to acute lung injury". Comment 2 Another concern is that the author describes the context in an engineering way that introduces several formulae, I think the author needs to add some contents from the biological side, such as what biological pathways are affected during the progression of lung fibrosis because of the pulmonary stretch and lung mechanotransduction. Answer to comment 1 We thank the Reviewer for this comment. In the paragraph entitled "The mechanotransduction process: biological response to stretch and progression in the fibrotic lung" we have summarized the main biological pathways implied in mechanotrasduction in lung fibrosis as reported in available literature. Minor comments Comment 1 In line 44, there is an ‘at’, please check and correct Answer to comment 1 We thank the Reviewer for this comment. We have corrected the manuscript. Comment 2 In line 163, Hook’s law did not show properly, please correct. Answer to comment 2 We thank the Reviewer for this comment. We have corrected the manuscript. Comment 3 In line 136, 137,148, 214, 221, what is that symbol: , please check throughout you whole text Answer to comment 2 We thank the Reviewer for this comment. We have modified the manuscript by replacing with the proper Greek's alphabet letter.